# Structure, Function and Engineering of the Nonribosomal Peptide Synthetase Condensation Domain

**DOI:** 10.3390/ijms252111774

**Published:** 2024-11-01

**Authors:** Zhenkuai Huang, Zijing Peng, Mengli Zhang, Xinhai Li, Xiaoting Qiu

**Affiliations:** College of Food Science and Engineering, Ningbo University, Ningbo 315800, China; hzhenkuai@163.com (Z.H.); 216003328@nbu.edu.cn (Z.P.); 18096770590@163.com (M.Z.); l15053937565@163.com (X.L.)

**Keywords:** nonribosomal peptide synthetase condensation domain, functional diversity of the condensation domains, nonribosomal peptide synthetase engineering targeting the condensation domain

## Abstract

The nonribosomal peptide synthetase (NRPS) is a highly precise molecular assembly machinery for synthesizing structurally diverse peptides, which have broad medicinal applications. Withinthe NRPS, the condensation (C) domain is a core catalytic domain responsible for the formation of amide bonds between individual monomer residues during peptide elongation. This review summarizes various aspects of the C domain, including its structural characteristics, catalytic mechanisms, substrate specificity, substrate gating function, and auxiliary functions. Moreover, through case analyses of the NRPS engineering targeting the C domains, the vast potential of the C domain in the combinatorial biosynthesis of peptide natural product derivatives is demonstrated.

## 1. Introduction

A number of approved pharmaceuticals are sourced from natural products or their derivatives [1]. Among these bioactive compounds, nonribosomal peptides (NRPs) represent a highly diverse category in terms of both functions and structures. NRPs exhibit a wide range of biological activities and can act as antibiotics, immunosuppressants, anticancer agents, metal carriers, pigments, and many other types of functional substances [2]. Diverse structures, including linear, cyclic, or branched forms, have been found in NRPs, which can be further modified to yield different chemical structures with multiple physicochemical properties [3] (Figure 1). NRPs are synthesized by large multi-modular megaenzymes referred to as nonribosomal peptide synthetases (NRPSs) [4]. NRPSs are among the largest and most complicated enzymes in nature, with their immense size due to their architecture, which is an assembly line consisting of a series of repeating catalytic units, referred to as modules, that are responsible for the incorporation of certain amino acids, or other types of acyl monomers in either a collinear or nonlinear mode [5,6]. In general, the number and arrangement order of modules corresponding to different monomer substrates directly correspond to the number and sequence of the peptide product [7]. In addition to the initiation module, each module typically contains at least three essential core domains: the condensation (C) domain, the adenylation (A) domain, and the peptidyl carrier protein (PCP) domain (Figure 2A).

The biosynthesis of NRPs is generally triggered by the specific adenylation of acyl monomer substrate by the A domain in the initiation module to form an acyl-AMP intermediate. Subsequently, the free thiol group of the 4′-phosphopantetheine (Ppant) arm of the PCP domain forms a thioester bond with the adenylated carboxyl terminal of the substrate, thus covalently linking the activated substrate to the PCP domain (Figure 2B) [8,9]. The PCP domain from the upstream module transports the loaded substrate to the active site of the C domain in the downstream module, and a peptide bond is formed, thereby transferring the upstream “donor” acyl monomer/peptide onto the downstream “acceptor” acyl-PCP, resulting in the peptide chain elongation (Figure 2C) [10]. Finally, there must be a domain responsible for the release of the mature product peptide chain covalently attached to the PCP domain in the termination module. This function is typically performed by the thioesterase (TE) domain present in the C-terminal of the termination module of the entire assembly line by catalyzing the release of mature peptide products through hydrolysis or cyclization [11]. In addition to these core functional domains, NRPSs selectively contain several types of auxiliary domains, such as methylation (M) domains that methylate the main chain amine and side chain groups of amino acid residues, oxidation (Ox) domains that oxidize the side chain groups of amino acid residues, and reduction (R) domains that stereo-specifically reduce carbonyl groups [12]. The presence of these multifunctional domains and their intricate coordination action modes results in the generation of diverse structures and wide biological activities of NRPs.

One of the goals of natural product research is to transform them into derivatives that are more suitable for human use such as therapeutical applications. Due to the complexity of these structures, NRPs are always difficult to be synthesized using regular chemical methods. Nevertheless, the collinear assembly line mechanism of biosynthesis employed by NRPS provides the possibility for engineering to produce peptide derivatives [13]. Although significant progress has been made in the engineering of NRPS systems over the years with many successful cases, engineered NRPSs often lead to reduced yields of the products, suggesting that our understanding of the NRPS biosynthetic mechanism is yet to be deepened. As one of the core functional domains of NRPS, the C domain profoundly influences the outcomes of engineering. Therefore, the study of C domain is an important aspect in the field of NRPS mechanism and engineering. In this review, the structure, mechanism, substrate selectivity, auxiliary functions of the C domain and NRPS engineering strategies targeting the C domain are comprehensively summarized. Moreover, it is noteworthy that there is currently no consensus on the mechanism of substrate selectivity of the C domain, which is discussed in detail as well.

## 2. Structural Characterization and Primary Function of the C Domain

### 2.1. The Overall Structure and Conformational Change in the C Domain

The structural study of the C domain has been extensively conducted. The first crystal structure reported for the C domain is VibH belonging to the vibriobactin synthetase system from *Vibrio cholerae* [14]. Dissimilar to most C domains, VibH is a standalone domain. Structures of the cores of the C domain are mostly conserved, providing a foundation for the study of homologous C domains catalyzing similar reactions. The C domain typically consists of approximately 450 amino acids and adopts a pseudo-dimeric chloramphenicol acetyltransferase (CAT) fold, composed of two subdomains at the N- and C-termini. Each subdomain consists of a small peripheral β-strand flanked by five α-helices. The two subdomains are connected by α-helices and interact more closely on one side, resulting in the overall V-shaped structure (Figure 3). In the C-terminal subdomain, a loop, known as the “latch”, extends to the N-terminal subdomain, followed by one or two loops at the end of the N-terminal β-strands (Figure 3). In addition to the gatekeeper, there is a second smaller crossover region called the “floor loop” [15,16], which stretches from the C-terminal subdomain to link with an α-helix at the N-terminal subdomain. Between the interfaces of the two subdomains, a cleft is formed. The donor and acceptor Ppant arms, tethered on the PCP domains, must cross this cleft from opposite sides to approach the conserved active site motif HHxxxDG, which is located in a loop connecting the central β-strand and a long α-helix of the N-terminal subdomain [17] (Figure 3).

There is limited information available regarding conformational changes in the C domain. Samel et al. proposed that the latch located above the conserved motif HHxxxDG needs to be opened to accommodate the substrate loaded on the Ppant arm of the PCP domain to access the key active site residues [18]. However, subsequent examples from different NRPS C domains show very limited relative orientation changes between the two subdomains, suggesting that these conformations are unrelated to the opening of the latch above the catalytic site [19]. The C domain exhibits variable relative arrangement between the N-terminal and C-terminal subdomains. This structural variability is believed to allow the domain to transition between “open” and “closed” states during various functional stages of NRPS biosynthesis [20]. However, Chu et al. found that the C domain of AmbB, a NRPS module involved in biosynthesis of L-2-amino-4-methoxy-trans-3-butenoic acid (AMB) in *Pseudomonas aeruginosa*, does not undergo significant conformational changes upon binding to the Ppant arm, as observed through comparisons with the PCP domain in three states (without Ppant, with Ppant, and with Ppant bound to the donor substrate L-Ala). Additionally, no significant conformational changes in this C domain were observed during the substrate delivery process [21].

### 2.2. Catalytic Mechanism and Substrate Selectivity of the C Domain

There are still some controversial issues that are relevant to the C domain, especially its catalytic mechanism and substrate specificity. The C domain catalyzes the formation of an amide bond, which is an essential step in NRP biosynthesis [18]. However, to date, the detailed catalytic mechanism of the C domain remains incompletely understood, and the precise catalytic roles of residues within the active site remain a topic of debate because of the catalytic function of particular histidine residues within the conserved active site motif HHxxxDG significantly varies among different C domains. Initially, it was believed that the key to the catalytic mechanism of the C domain lies in the second histidine within this conserved motif. This histidine residue was proposed to act as a catalytic base, deprotonating the α-amino group of the acceptor aminoacyl-PCP to allow for the nucleophilic attack on the carbonyl group of the donor peptide-bound PCP thioester [17,22]. Alternatively, it acts as a general base to stabilize the tetrahedral transition state [18,23]. However, this role is not absolute, as the mutation of this residue does not always abolish catalytic activity in certain C domains [14,24]. Theoretical estimates based on pK_a_ values suggest that the second histidine residue in the “HHxxxDG” motif is protonated under physiological conditions, which contradicts its role as a general base [18]. In addition, some researchers have proposed another mechanism where the role of this active site histidine is to position the α-amino group of the acceptor substrate for facilitating its nucleophilic attack as well as to determine the acceptor substrate specificity [25,26].

In the working mechanism of NRPS, the A domain primarily controls substrate selection, and the C domain also plays a role as a secondary gatekeeper for substrate specificity. In cases where the A domain incorrectly selects the substrate, the C domain can perform a second substrate proofreading, thereby significantly reducing the error rate in monomer incorporation. The acceptor site of the C domain is considered to be its primary selection filter in this process [27]. Belshaw et al. loaded the chemically synthesized aminoacyl coenzyme A molecules via the broad-specificity phosphopantetheinyl transferase Sfp in vitro to prepare PCP domains carrying various aminoacyl groups [28]. This method bypasses the substrate selection by the A domain while allowing for the investigation of substrate selectivity of the C domain, including both donor and acceptor sites. The results indicate that the acceptor site of the C domain exhibits extremely high selectivity in both side-chain structure and stereoconfiguration of α-carbon, as it can accept homologous L-propionyl-PCP and non-homologous L-alanyl-PCP, while D-alanyl-PCP, L-leucyl-PCP, and D/L-phenylalanyl-PCP cannot. In contrast, the selectivity of the donor site is lower, as both homologous L-propionyl-PCP and non-homologous L-phenylalanyl-, L-alanyl-, D-alanyl-, or L-leucyl-PCP can serve as donors [28]. Based on the stereochemical selectivity of amino acid substrates, the C domain can be divided into ^L^C_L_ and ^D^C_L_ domains: the ^L^C_L_ domain primarily mediates the condensation of two L-amino acids, while the ^D^C_L_ domain catalyzes the condensation of an upstream D-amino acid and a downstream L-amino acid [29]. Subsequent characterization of C domain selectivity was performed in NRPS responsible for glycopeptide antibiotic biosynthesis, revealing that before the entrance of substrates into the condensation domain, two crucial modifications need to occur. The role of the C domain is to ensure that the aminoacyl thioester tethered in the PCP domain is in the correct modification state before condensation occurs [30]. The structure of the C domain in complex with the aminoacyl-PCP acceptor substrate, as revealed by Izoré et al., provides an excellent model for understanding substrate selection by the C domain. This study found that the channel through which the Ppant arm enters the active site of the C domain appears to be regulated by the residue R2577, which is highly conserved in the ^L^C_L_ domain and is typically occupied by glycine or other small side-chain residue in the corresponding site of ^D^C_L_ domain. This residue repels unmodified Ppant arms (or neutral/negatively charged substrates) and tends to favor aminoacyl-Ppant [31]. Meanwhile, some researchers have raised questions about the substrate specificity of the C domain. Calcott et al. argue that the substrate selection of the C domain is an exception rather than a general rule. They completed the previously unfinished replacement of the A domain to generate modified pyoverdine and found that the C domain in the *Pseudomonas aeruginosa* peptidyl NRPS system does not impose strict substrate proofreading constraints. Additionally, they discovered that in PheATE/ProCAT bimodular NRPS synthesizing tyrocidine, leucine could also serve as the acceptor residue. Previous studies indicated that at this position, this bimodular NRPS originally did not accept artificially loaded leucine thioesters, which was the basis of the initial assumption that the C domain exhibited extremely high selectivity for acceptor substrates [32]. Schopp et al. also pointed out that the C domain exhibits broad substrate selectivity and tolerance towards different stereochemical structures [10]. Moreover, during the investigation of product specificity of bimodular NRPS sdV-GrsA/GrsB1 under substrate competition through kinetic modeling, Stanišić et al. demonstrated thatsubstrate specificity of C domain can be overcome by A domain activity [33].

## 3. The Auxiliary Functions of the C Domain

The canonical C domains enable the elongation of peptide chains by catalyzing condensation reactions between substrates tethered to the PCP domains from adjacent modules [34]. In addition to its conventional functions, the C domain exhibits a highly diverse range of auxiliary functions during NRP synthesis. Rausch et al. classified the subtypes and superfamilies of C domains based on their catalytic properties into ^L^C_L_ subtype (catalyzing the condensation between L-aminoacyl/peptidyl-PCP donor and L-aminoacyl-PCP acceptor), ^D^C_L_ subtype (catalyzing the condensation between D-aminoacyl/peptidyl-PCP donor and L-aminoacyl-PCP acceptor), Starter C (C_S_) domain, as well as non-canonical C superfamily members including epimerization (E) domain and dual E/C domains, heterocyclization (Cy) domain, X domain, and condensation-like or termination condensation (C_T_) domain [33]. Details of some of the auxiliary functions of the C domain are discussed below.

### 3.1. Starter C Domain

Typically, the initiation module of NRPS lacks the C domain because substrate loading in the initiation module does not require a condensation reaction. However, in the NRPSs for synthesizing certain lipopeptides such as surfactin, lichenysin, fengycin, and arthrofactin, the C domain is found as the first domain in the initiation module, referred to as C_S_ domains [35]. These lipopeptides listed above share a common feature: the first amino acid in the peptide chain is linked to a β-hydroxyl fatty acid. Therefore, the C_S_ domain is also considered to be an acceptor for the fatty acids delivered from acyl transferases [36]. In addition to fatty acids, other types of monomers can also be linked to amino acids through C_S_ domains. For example, in *Kibdelosporangium* sp. AK-AA56, during the synthesis of the heptapeptide antibiotics JBIR-78 and JBIR-95, the C_S_ domain catalyzes the condensation of phenylacetyl-CoA with the N-terminal L-Val tethered to the PCP domain [37].

### 3.2. Epimerization Domain

The E domain, belonging to the C domain superfamily, also contains a conserved HHxxxDG motif [17] and was discovered during the initial investigation of the C domain. This domain can epimerize the configuration of the α-carbon of substrate monomers from L to D-configuration [4]. Dissimilar to ribosomal peptides, NRPs frequently contain D-amino acids, which can attenuate the action of L-configuration-specific proteases, thereby notably slowing down the degradation of NRPS products. Additionally, D-amino acid–containing NRPs enable specific conformations, which are necessary for further processing or biological activity [38,39]. In the downstream region from the E domain, there is always a ^D^C_L_-type C domain, as the E domain catalyzes the conversion of configuration in a non-specific manner, resulting in a mixture of substrates with both configurations [35]. Through selective screen, this C domain can ensure the correct stereoisomer is utilized in subsequent extension reactions. Moreover, in several cases, this process is performed by dual E/C domains [40].

### 3.3. Heterocyclization Domain

In several particular modules of NRPS, the C domain may be replaced by a Cy domain. Although it shares overall structural similarity with the canonical C domain, the Cy domain contains a conserved DxxxxD motif instead of the HHxxxDG motif generally found in the C domain [41,42] (Figure 4). This type of C domain is initially discovered in bacitracin synthetase [42]. The Cy domain can catalyze two consecutive reactions: initially, it catalyzes the formation of an amide bond; subsequently, it catalyzes an intramolecular dehydration between the side chain and the carbonyl carbon of the main chain of amino acid substrates such as Cys, Ser, or Thr, forming a thiazoline, oxazoline, or methyloxazoline ring, respectively. These rings are crucial for the structural stability and function of the peptide product [41]. The generated heterocycles can undergo further reduction or oxidation by reductase (R) or oxidase (Ox) domains, respectively [43,44].

### 3.4. X Domain

Glycopeptide antibiotics (GPAs) are a class of cell wall biosynthesis inhibitors with a unique mechanism of action involving the binding of substrates involved in cell wall biosynthesis, thereby inhibiting the growth of Gram-positive bacteria [45]. Oxidative cross-linking of aromatic side chains catalyzed by cytochrome P450 monooxygenase is required for GPAs to achieve their final active conformation [46]. The X domain is a unique recruitment domain involved in the biosynthesis of GPA. This domain is responsible for recruiting multiple cytochrome P450 monooxygenases to the NRPS-bound peptide, facilitating the necessary side chain cross-linking for peptide oxidation or cyclization [46,47,48]. The X domain adopts a C/E-type folding, characterized by a V-shaped arrangement of its two subdomains, both of which belong to the CAT fold with the region corresponding to canonical C-domain active site features the HRxxxDD motif, which is located in the N-terminal subdomain (Figure 5) [46].

### 3.5. C Domain Responsible for Substrate Release

As mentioned above, in canonical NRPS assembly lines, the catalysis for the release of the final product is typically performed by the TE domain. However, not all assembly lines rely solely on TE domains for this function. In certain NRPS systems, the C domain at the end of the assembly line is also involved in various types of substrate release. The C domain can facilitate substrate cyclization through multiple mechanisms. For example, at the terminal end of many fungal NRPS assembly lines, there exists a C_T_ domain, which catalyzes the macrocyclization by using the terminal amine as a nucleophilic agent during the chain release reaction (Figure 6) [49]. Although C_T_ domains share the conserved HXXXDXXS motif with canonical C domains [49], structural characterization of C_T_ domain revealed several structural features that are distinct from those of canonical C domains, especially the tight interaction between the two subdomains of C_T_ domain near the acceptor side, leading to the blockage of the solvent channel for accommodating acceptor substrate (Figure 7), which is consistent with the fact that no downstream acceptor substrate is required for cyclization of the donor substratecatalyzed by C_T_ domain [50]. Compared to cyclization, there are fewer examples of C domains releasing products through hydrolysis. A special single-module NRPS from *Aspergillus nidulans*, referred to as IvoA, is associated with pigment biosynthesis [51]. In IvoA, the A domain activates and transfers L-tyrosine to the PCP domain, the E domain subsequently catalyzes the stereoinversion of the substrate, and the C_T_ domain finally selectively hydrolyzes D-tyrosyl-S-phosphopantetheine thioester to release D-tyrosine.

In addition to these non-canonical functions described above, C domains in NRPS assembly lines play various roles, such as ester bond formation, isopeptide bond formation, β-lactam formation, Pictet–Spengler cyclization, etc. [52]. These diverse functions of C domains not only deepen our understanding of the mechanism of NRPS but also provide a basis for the development of biocatalytic tools for synthesizing novel derivatives of peptide natural products with broad application prospects.

## 4. Engineering of the C Domain

### 4.1. The Substrate Specificity of the C Domain and Its Significance in Engineering

The unique mechanism of NRPS and its modular assembly of acyl monomers have provided significant opportunities for engineering. Since the modular combination principle of NRPS was discovered, it has inspired scientists to develop synthetic biology approaches to rationally or randomly construct hybrid NRPS. The ultimate goal of NRPS engineering is to obtain customized products, so particular attention has long been paid to the A domain when developing strategies for NRPS engineering. However, despite the apparently independent action mode of NRPS modules and domains, there are coordinated connections between them. As discussed above, not only can the A domain specifically select the substrates, but the C domain can also exhibit strict substrate specificity. After the A domain selectively recognizes and adenylates the substrate, the C domain can further verify the acceptor substrate, ensuring the specificity of the sequence of the synthesized peptide product.

The sophisticated architecture of NRPS, while advantageous for its functionality, significantly complicates its engineering. When attempting to alter only the substrate specificity of the A domain, the C domain can become a potential barrier [29]; the final products obtained from such engineering are always prone to deviate from designed targets [53,54]. The ability to achieve substrate specificity changes is usually limited to substrate side chain selection by the A domain, and such alterations often come with a significant decrease in catalytic efficiency, as the C domain cannot accommodate substantial changes in the side chain structure of the acceptor substrate. In general, the A domain can only tolerate minor alterations, allowing for the loading of amino acids with similar side-chain structures. Otherwise, it would severely affect the synthesis efficiency of NRPS [55]. On the other hand, there are reports indicating that the C domain can influence the substrate specificity of its adjacent A domain. During the investigation of the NRPS mechanism of microcystin, a class of cyanobacterial hepatotoxins, Mayer et al. demonstrated that when the McyC A domain is expressed in the absence of C domain, it can adenylate a variety of amino acids, but this A domain only exhibits adenylating activity towards Arg when co-expressed with the C domain [56]. They also found that the substrate specificity of the McyB A domain is strongly shifted to Leu when co-expressed with the C domain, while it can accept various hydrophobic side-chain amino acids, such as Leu, Val, Ile, and Tyr, as substrates when expressed in the absence of C domain. Moreover, the same effect for the restriction of substrate profile of the A domain was also observed when only the C-terminal subdomain (C_CTD_), which directly interacts with the A domain, was co-expressed [56]. These observations illustrate that the interaction between the C domain and A domain is capable of impacting the substrate adenylation specificity of the A domain, and the C domain can also influence the activity of certain A domains.

The examples described above demonstrate the importance of the C domain in the substrate specificity of NRPS, which is pivotal for the development of strategies for NRPS engineering.

### 4.2. C Domain Replacement and Swapping

In some particular types of fungal NRPSs, there are three distinct C domains: a non-catalytic N-terminal C domain (C1), a classical C domain (C2) catalyzing amide bond formation, and a C-terminal C domain (C3). C3 is a critical domain that possesses three functions, including condensation, chain length control, and macrocyclization. In the biosynthetic pathways of enniatin, beauvericin, and bassianolide, the C-terminal ester bond formation catalyzed by the C3 domain controls the chain length of the final products. By swapping these C3 domains or PCP-C3 bidomains, products with different chain lengths can be generated [57]. Initially, expression of these engineering modules was performed in *Escherichia coli*, but because of the limited acceptance of this type of exchange by the bacterial NRPS system, although the target products were obtained, the yield was relatively low. Subsequently, using *Aspergillus niger* as a host for heterologous expression of PCP-C3 bidomain exchanged NRPS, the production of octa-enniatin B, octa-beauvericin, and hexa-bassianolide significantly increased [58]. These observations herald the potential of fungi as hosts for heterologous expression of engineered NRPS.

In the biosynthesis of fumiquinazolines and tryptoquialanine, the single-modular NRPSs Af12050 and TqaB select and activate either alanine or isobutyric acid, which are then condensed with oxidized fumiquinazoline F. The generated imidazoindolone scaffolds exhibit differences in stereochemistry at the C-N bond, which is determined by the C-terminal C domain of NRPS. By swapping these highly homologous C domains, stereochemical inversion of these peptides was successfully achieved in vitro [59].

In the biosynthesis of streptothricin (ST) antibiotics, a gene cluster associated with ST biosynthesis that includes 24 ORFs was investigated [60]. Four ORFs that encode NRPS were identified in this gene cluster: ORF5 encoding a standalone A domain, ORF13 encoding A and PCP domains, ORF18 encoding C and PCP domains, and ORF19 encoding a standalone A domain. The C domain encoded by ORF18 catalyzes the formation of intermolecular amide bonds between PCP-bound oligopeptide thioester containing L-β-lysine and aminoglycoside streptothrisamine. It has been demonstrated that the catalytic efficiency of ORF18 affects the length of the final product. The C domain of ORF18 possesses the HQxxxDM motif instead of the common HHxxxDG motif, and mutation of the glutamine residue at the active site to alanine reduces the enzyme’s catalytic activity, leading to the production of oligopeptides containing L-β-lysine with varying lengths in vitro, some of which cannot be produced by the wild-type enzyme.

The in vitro analysis of the C_s_ domain of NRPS that biosynthesizes the lipopeptide A54145 has identified candidate amino acid residues that may interact with the acyl chain in the substrate channel. To assess the impact of these residues on substrate selection, a series of C_s_ domain mutants were constructed, and their effects were biochemically evaluated. Although some mutants abolish the catalytic activity, others showed changes in substrate specificity towards shorter or longer acyl chains [61].

In a previous study, researchers reprogrammed the C domain to alter its substrate specificity. They conducted high-throughput analyses of various C domains for selecting the appropriate one as the engineering target: the last C domain of a surfactin synthetase was reprogrammed to be able to accept fatty acid substrates instead of amino acids for enhancing antimicrobial activity [62]. Researchers chose to engineer SrfA-C, the microbial lipopeptide antibiotic surfactin synthetase while employing a designed variant of TycA as the upstream module (Figure 8) [63]. RzmA-C_s_ and HolA-C_s_ domains are the C_s_ domains responsible for initiating the biosynthesis of lipopeptidesrhizomideand holrhizinthrough the acylation of leucine and valine with short-chain fatty acids, respectively [26]. The acyl chains of both of these two lipopeptides can be changed by swapping the RzmA-C_s_ and HolA-C_s_ domains or utilizing mutants of these two C_s_ domains generated by site-directed mutation of a key residue responsible for controlling the acyl chain length [26], which is similar to the engineering case of C_s_ domain for A54145 demonstrated above. Moreover, the donor substrate can be extended up to palmitoyl-CoA from native acetyl-CoA through additional point mutations of several key residues that determine the selectivity of lipid chain length characterized through the structural analysis of octanoyl-CoA binding site of this RzmA-C_s_ mutant [26]. Through comparison of the structure of the C domain of SrfA-C with that of RzmA-C_s_, it was observed that an expansion of the substrate pocket between the core β-fold sheet and the fourth α-helix of the C domain of SrfA-C could potentially enhance the recognition of 10-undecynoic acid. A series of hydrophobic residues were identified based on structural comparisons and were subjected to site-directed mutagenesis. Among these mutants, SrfA-C harboring mutations W143T, Y145V, and F155I exhibited rapid acylation of L-Leu, with activity levels surpassing that of the wild-type by up to 40-fold (Figure 8) [64].

Kaniusaite et al. conducted inter-module domain swapping in the biosynthetic pathway of the glycopeptide antibiotic teicoplanin. By relocating the Edomain, they enhanced the racemization capacity of several certain modules. For instance, replacing the E domain of the M4 module with that of the M5 module improved the accuracy and yield of peptide products. Additionally, they redesigned the cross-module communication-mediating domain, which will be introduced in detail in Section 4.4, to ensure effective communication between the separated NRPS modules, thereby enabling cross-module peptide chain assembly [65].

### 4.3. Multi-Domain Replacement Involving C Domain

Since the engineering of individual A domains proved to be challenging in overcoming the influence of the C domain, researchers have explored the replacement engineering of NRPS assembly lines by employing multiple domains as interchangeable units while maintaining homology between the C and A domains. Preserving homology between the C and A domains not only reduces the repulsion of the C domain toward the A domain but also expands the substrate specificity of the C domain for upstream donor sites, thus bypassing the substrate specificity of the C domain more effectively. Marahiel’s team combined modules 2 with modules 9 or 10 of the tyrocidine synthetases to construct bimodular artificial subunits C-A-PCP_Pro_-C-A-PCP-TE_Leu_ and C-A-PCP_Pro_-C-A-PCP-TE_Orn_ (Orn: ornithine). When co-incubated with the initiation module, both constructs were able to synthesize tripeptide products consistent with expectations, indicating that recombination using C-A-PCP as interchangeable units can yield functional hybrid NRPS [66]. The PvdD derived from *Pseudomonas aeruginosa* PAO1 is a bimodular NRPS responsible for linking the final two serine residues into the peptide chain. Calcott et al. initially conducted domain replacement experiments on the first module of PvdD and found that substitutions with the C-A bidomain were non-functional [53]. Subsequently, they replaced the C-A bidomain of the second module of PvdD with a series of homologous or heterologous C-A bidomains, thereby avoiding the potential disruption of critical interactions between PvdD and PvdJ in the first module. The results revealed that two hybrid NRPSs, C-A_Thr_-A and C-A_Lys_-C, exhibited better catalytic activity, with product yields of 83% and 76% of that of the wild-type, respectively, while the remaining six hybrid enzymes showed significantly reduced activity or were abolished. Although multi-domain replacement generally works, both the success rate and conversion efficiency were not extraordinary [54].

Recently, a team led by Bode from Germany achieved significant advancements and findings in the field of multi-domain replacement. They propose that the linker region between the C and A domains serves as an ideal fusion site for domain exchange. In contrast to the linker regions of A-PCP and C-PCP domains, the linker region of C-A domains is longer and more flexible, along with a number of specific interactions between these two domains [18,67]. Researchers split different NRPSs at the linker region of the C-A domains for obtaining a series of A-PCP-C or A-PCP-C/E exchange units. These exchange units, denoted as XUs, can be properly replaced or recombined to construct hybrid NRPSs (Figure 9). To validate the feasibility of constructing hybrid NRPSs using XUs, the XU of ambactin synthetase AmbS was replaced with XUs from a variety of NRPS modules possessing similar substrate specificities [68]. The results showed that one hybrid enzyme among these constructs generated the desired product with nearly unchanged activity. Moreover, another hybrid enzyme, characterized by the inclusion of the E domain in the replaced XU, demonstrated the ability to generate substrate derivatives with a yield that was 48% higher than that of the wild-type (Figure 10A; Table 1). Subsequently, researchers obtained a series of non-natural NRPs through the recombination of NRPSs and the de novo construction of NRPSs (Figure 10B–E) [68]. However, this method also has limitations. First, it relies on the presence of a conserved WNATE sequence in the linker region between the C and A domains, which is not found in many proteins, greatly reducing the universality of this approach. Second, it is essential to ensure that the substrate recognized by the hybrid A domain matches the substrate specificity of the C domain in the original module. Otherwise, the substrate may not be able to enter the C domain, leading to the failure of peptide chain elongation.

The C domain is typically divided into the donor subdomain (CD_sub_) and the acceptor subdomain (CA_sub_). Bode’s team developed a strategy termed exchange unit condensation domain (XUC), where CA_sub_-A-PCP-CD_sub_ serves as a replacement unit for overall domain swapping [69]. This approach places the A domain and CA_sub_ within the same exchange unit, thus circumventing the issue of incompatible substrate specificity between the hybrid A domain and the C domain, so this method offers great universality. Experimental results demonstrated that peptides generated using this approach showed a 63.4% increase in yield compared to that of the wild-type, confirming the feasibility of this strategy [64]. Another approach involves the utilization of a special synthetic peptide known as synthetic zippers (SZs) [70]. SZs are capable of achieving specific binding between two proteins through their coiled structure with high affinity [71]. Researchers divided the XU into two subunits at the C-A junction and then utilized the WNATE motif between the C and A domains as an insertion position to combine the two subunits using SZs. This combination of NRPSs is termed S-type NRPSs, and some S-type NRPSs even exhibited a yield that is up to nine times greater in comparison to that of the wild-type. In subsequent work, researchers extended this concept by applying SZs to C-PCP and A-PCP domains, resulting in the generation of 34 recombinant S-type NRPSs and the production of 47 unique peptides [72].

The sources for NRPS engineering have extended beyond linear NRPS to include various types of NRPSs, such as iterative NRPSs that are capable of reusing single modules or even the whole synthetase. Steiniger et al. combined linear and iterative NRPSs by utilizing three exchange units, C-A-Mt-PCP, C_CTD_-A-Mt-PCP, and A-Mt-PCP (Mt: *N*-methyltransferase domain), derived from three distinct fungi, to generate 24 hybrid synthetases, some of which yielded novel peptide products. The observations of this engineering indicated that the maintenance of the specificity and integrity of the C domain is crucial for achieving functional recombination. This novel strategy of combining different assembly modes of NRPS opens up new avenues for the production of novel bioactive peptides [73].

### 4.4. Engineering of COM Domain

During the engineering of NRPS, in addition to considering the conformational changes within domains, attention should also be paid to the interactions between domains and between modules. The communication-mediating (COM) domain plays a crucial role in facilitating proper communication between collaborating NRPS components, preventing undesirable interaction between uncoupling components, thus ensuring the assembly of peptide products according to specific template requirements [74].

The COM domain, also known as the docking domain (DD), was initially thought to exist between the upstream E domain and the downstream C domain within a module. However, subsequent research revealed the presence of COM domains between domains within modules [75,76]. The location of COM domains within modules is illustrated in Figure 11. The COM domain can be divided into the N-terminal acceptor domain (COMA domain) and the C-terminal donor domain (COMD domain). Within the E domain, the COM domain contains a highly conserved sequence, TPSD, which serves as a fusion site between the E domain and the C-terminal COMD domain. Additionally, the N-terminal COMA domain also harbors a conserved sequence, L(T/S)P(M/L)QEG, which functions as a fusion site between the C domain and the COMA domain [76].

Stachelhaus et al. were the first to discover a COM domain comprising 15–30 amino acids between partner subunits (TycA, TycB, TycC) of tyrocidine synthetase from *Bacillus brevis* (Figure 11A) [75]. To validate its impact on inter-subunit interactions, researchers mutated the C-terminus of subunit TycA (COMD domain) while maintaining the functionality of the A and E domains. Mutants with mutations spanning residues 6–23 at the C-terminus failed to produce the expected products. For mutations in the N-terminal COMA domain, since the activity of the isolated C domain was difficult to determine, researchers assessed the effect of the absence of the COM domain at the N-terminus on product generation by exchanging matched and incompatible COM domains. Results showed that the product could be generated only when compatible COM domains were present, whereas no product was generated when they were replaced with non-matched COM domains, suggesting that both COMD and COMA domains, as well as their compatibility, play crucial roles in facilitating communication between the two partner subunits.

Plipastatin is a cyclic lipopeptide antibiotic synthesized by a synthetase composed of five subunits: PpsA, PpsB, PpsC, PpsD, and PpsE. COM domains are also discovered between these subunits (Figure 11B). Gao et al. generate engineered NRPSs by moving the N-terminus containing the natural COMA domain of the subunit PpsE forward to the C-termini of the subunits PpsB and PpsC containing the natural COMD domain, respectively (referred to as LP7 and LP8) [76]. LP7 produced a novel cyclic pentapeptide, which exhibited potent antibacterial activity against *Rhizopus stolonifer* and *Staphylococcus,* even stronger than that of plipastatin. However, LP8 did not yield any products. Subsequently, researchers generate engineered NRPSs by replacing the incompatible COMD-PpsC/COMA-PpsE with the naturally compatible COMD-PpsC/COMA-PpsD and COMD-PpsD/COMA-PpsE, respectively (referred to as LP9 and LP10). Product formation was observed in both engineering strains. LP9 produced a linear hexapeptide instead of the expected heptapeptide, indicating that the replacement of COMD-PpsC/COMA-PpsD partially aided PpsA, PpsB, PpsC, and PpsE in restoring its substrate synthesis capability but failed to guide substrate condensation occurred between the two subunits. By contrast, LP10 yielded the expected heptapeptide product. These observations illustrate the importance of COM domain compatibility and the crucial role of the COM domain in the facilitation of interaction between two subunits in maintaining NRPS functionality. Additionally, it is noteworthy that Hahn et al. proposed that the ability for selective interactions between subunits can be retained by replacing them with highly homologous COM domains [77]. However, the sequence homology between acceptor COMA-PpsC and COMA-PpsE is only 30%, demonstrating a broad range of options and flexibility when performing the replacement of COM domains. Furthermore, COM domains are not limited to being involved in the interactions between subunits. Through bioinformatics analysis, Fage et al. revealed that COM domains are widely present both between and within modules of NRPS [74]. For instance, Lv et al. discovered the presence of COM domains between domains in the bacillomycin D synthetase [78] (Figure 12A). They observed changes in product generation by deleting the COMD and COMA domains between three pairs of domains in bacillomycin D synthetase (Figure 12B–D). Strains lacking the COMD domains produced truncated peptides, and two mutant strains even produced bacillomycin D, with one strain producing bacillomycin D identical to the wild-type. In addition to detecting the expected products, the presence of truncated peptide products revealed the existence of a new assembly line for generating cyclic hexapeptides. The researchers suggested that the absence of COMD domain in BamB might lead to module skipping in the bacillomycin D assembly line, specifically the skipping of module 4, resulting in a new cyclic hexapeptide assembly line (Figure 12E), indicating that the absence of the COMA domain does not affect the catalytic activity of upstream modules but influences that of the downstream modules for peptide chain elongation. The enhanced flexibility in biosynthetic systems demonstrates potential applications of COM domains located between domains.

Comparative results listed above show a great impact of COMA domain deletion on the functions of NRPS assembly lines, indicating that COMA domains are crucial components for inter-module interactions and effective communication. Xefoampeptide synthetase (XfpS) is a relatively small three-module NRPS responsible for synthesizing XFP A and XFP B. Researchers tracked soluble XfpS or its derivatives by adding Strep Tag II tags at both N- and C-termini, leading to a significant increase in the yield of wild-type XFP. The three modules of XfpS are XfpS-1 consisting of C-A-PCP-E domains, XfpS-2 consisting of C-A-PCP domains, and XfpS-3 consisting of C-A-PCP-TE domains. XfpS was split into single modules and module pairs, and COM domains were introduced at the corresponding N- and C-termini (Table 2). The results showed that after splitting and introducing COM domains at the E-C domain junction of XfpS-1 and XfpS-2, XFP can still be efficiently synthesized. However, splitting at the PCP-C domain junction of XfpS-2 and XfpS-3 significantly affected their activity. Upon the complete splitting of XfpS and the introduction of two sets of COM domains, XFP production further decreased [79].

The examples listed above illustrate the potential of engineering the COM domain to mediate interactions between modules and to facilitate the production of diverse NRPs. In contrast to the extensive research on classic modules of NRPS, the study of the COM domain is still insufficient, with many issues yet to be explored.

## 5. Conclusions and Prospects

The structural analyses of a variety of C domains have provided deep functional and structural insights into this domain. Although we have gained some understanding of the catalytic mechanism, substrate specificity, and conformational changes during various catalytic statuses of C domains, major questions regarding their catalytic mechanism and substrate specificity still remain to be resolved. The debate surrounding C domains is mainly relevant to the exact function of the second histidine residue within the conserved active site motif and the general rules governing substrate amino acid specificity in both side-chain structure and stereochemistry. New perspectives are continuously being proposed: for example, Mansour et al. employed a multiscale computational approach to investigate the catalytic mechanism of the C domain of tyrocidine synthetase, revealing that the second histidine residue within the conserved active site motif (His222) initially acts as a base to facilitate nucleophilic attack of the prolyl nitrogen at the phenylalanyl’s carbonyl group for the formation of the amide bond, and the protonated His222 subsequently acts as an acid to protonate the thiolate and recovers to a neutral form [80]; Peng et al. identified three key mutations in C domain that govern the preference for side-chain structure and stereochemistry of substrate via extensive examination of the peptide formation of C domain mutants in a bimodular NRPS system [81]. Moreover, one of these mutations, from a glutamate to a leucine residue within the active pocket, leads to the stereopreference switch of the donor substrate from D- to L-configuration. Nevertheless, clear conclusions regarding these two issues have not yet been reached. Future research on C domains may provide definitive answers to these questions and introduce novel viewpoints. The discovery of noncanonical C domains and other C superfamily domains has also expanded our understanding of C domains, and these auxiliary domains have broad prospects for application in NRPS engineering. On the other hand, the ongoing discovery of C domains with additional functions also suggests that our exploration of the extra functionality of C domains is still yet to be explored. With technological advancements, C domains possessing more novel and specific functions will be expected to be discovered in the future.

Based on the linear catalytic mechanism of NRPS, directed recombination of catalytic modules can achieve the corresponding peptide sequences in principle. However, the catalytic efficiency of hybrid NRPS limits its feasibility and applicability. The substrate specificity of each module within NRPS mainly depends on the A domain and the C domain, making these two domains the focus of NRPS engineering. The importance of the A domain in directly selecting substrates to synthesize the corresponding aminoacyl-AMP is undeniable. However, the C domain, as the second threshold for substrate selection in the NRPS assembly line, also has a significant impact on the outcome of NRPS engineering by verifying the correctness of substrates. Current observations of NRPS engineering demonstrated above suggest that compared to the replacement of individual domains, the strategy of replacing the boundary between the C and A domains together with these domains is more advantageous. This strategy not only generates the target product but also can even elevate the yield to several times higher than that of the wild-type. These successful cases of engineering demonstrate that the selection of suitable target catalytic units and appropriate recombination boundaries enables hybrid NRPS to recognize and load diverse acyl monomer substrates while improving catalytic efficiency. Although there has been significant progress in NRPS engineering, there still lacks a universal strategy for stably producing target NRP products. This is mainly due to the incomplete understanding of the mechanism of action of the C domain and even the entire NRPS, which hinders the development of NRPS engineering. The future focus of NRPS studies should first be on the in-depth study of the mechanism of NRPS, especially of the substrate recognition mechanism of the C domain, to reveal more universal rules for developing more efficient and stable engineering strategies. The engineering approach should first ensure that the engineered NRPS retains complete catalytic activity and then aim to observe improved catalytic efficiency to generate the expected final product with a high yield. Moreover, the combination of different types of NRPSs, such as iterative, translocation, and module jumping modes, may better facilitate the development of NRPS engineering strategies: iterative and translocation NRPSs can aid in the synthesis of more complicated polypeptide structures, while module jumping NRPS allows for the design of novel synthetic pathways. Additionally, the introduction of non-natural amino acid residues or multiple modifications could lead to the development of products with new functionalities. Altogether, the ultimate goal of NRPS engineering is to rationally design construction strategies that allow for arbitrary recombination of catalytic units, thereby efficiently synthesizing customized peptide products, which is of great significance for increasing both the diversity of the NRP family and developing new drugs.

## Figures and Tables

**Figure 1 ijms-25-11774-f001:**
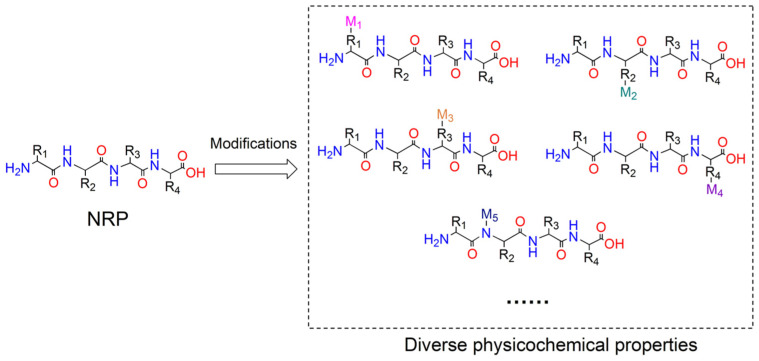
NRPs are capable of being modified by NRPS catalytic domains or tailoring enzymes to generate a variety of derivatives possessing diverse physicochemical properties. Abbreviations: R1-R4 stand for side chains of residues in NRP; M1-M5 stand for the modification groups.

**Figure 2 ijms-25-11774-f002:**
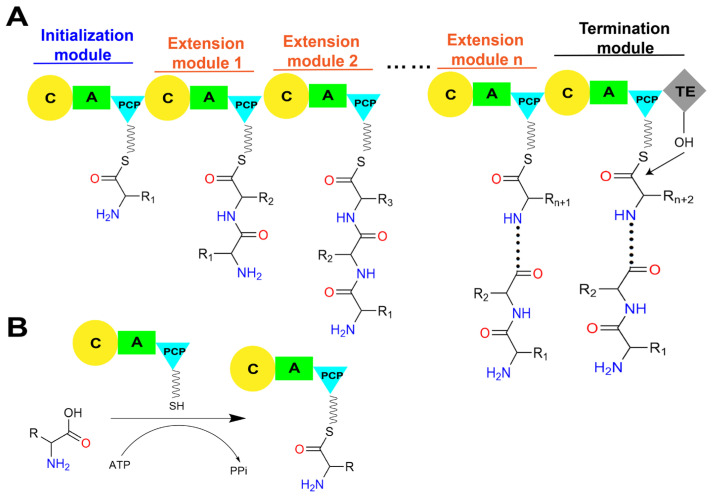
The linear assembly line in a typical NRPS and the primary roles of core catalytic domains. (**A**) The assemble of peptide chain by linear NRPS. (**B**) The A domain consumes ATP to activate amino acid substrates and load them onto the PCP domain to generate an aminoacyl-*S*-carrier protein complex. (**C**) The C domain within the NRPS module catalyzes peptide bond formation, linking the substrate to the growing peptide chain and presenting it to the downstream module. Abbreviations: A: adenylation domain; C: condensation domain; PCP: peptidyl carrier protein domain; TE: thioesterase domain; CoA: coenzyme A; 3′, 5′-ADP: 3′, 5′-adenosine diphosphate; ATP: adenosine triphosphate; PPi: pyrophosphate.

**Figure 3 ijms-25-11774-f003:**
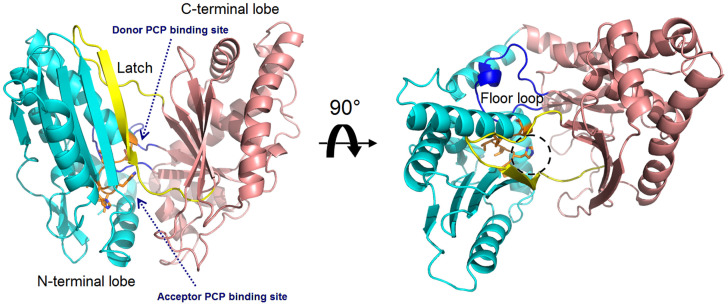
Overall structure of the VibH (PDB code: 1L5A). The N-terminal lobe is shown in cyan, while the C-terminal lobe in pink. The side chains in conserved HHxxxDG motif is represented as orange sticks, with the catalytic histidine residue marked by a dashed circle. The latch and floor loops are displayed in yellow and blue, respectively. The binding sites of the donor and acceptor Ppant arms tethered on the PCP domains are indicated.

**Figure 4 ijms-25-11774-f004:**
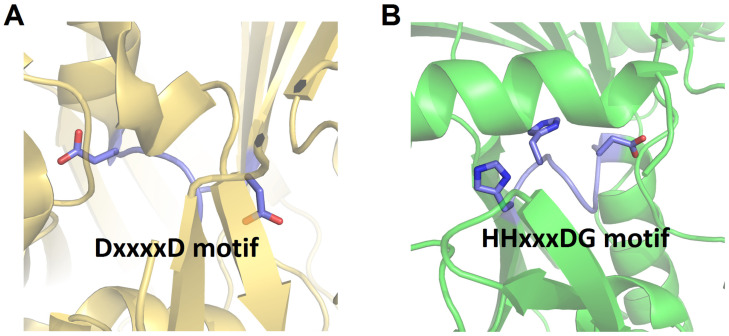
Structural comparison of the active sites of Cy domain of BmdB from *Thermoactinomyces vulgaris* (PDB code: 5T3E) (**A**) with canonical C domain (VibH) (**B**). The loop containing the key active residues is highlighted, and the side chains of the conserved residues within the active site motifs are represented as sticks.

**Figure 5 ijms-25-11774-f005:**
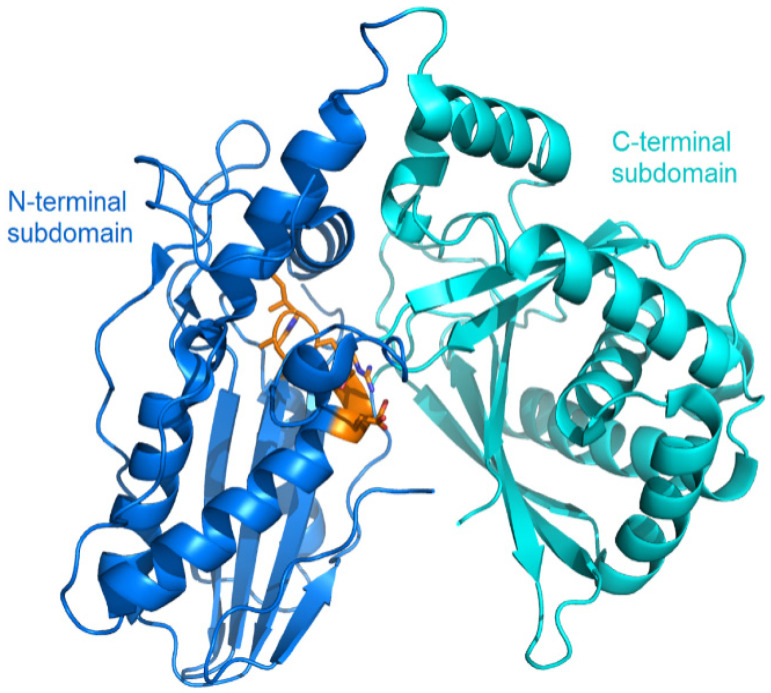
Overall structure of the X domain of Tcp12, the terminal module for biosynthesizing teicoplanin from *Actinoplanesteichomyceticus* (PDB code: 4TX2). The two subdomains are represented in different colors, and the side chains in the HRxxxDG motif (corresponding to the canonical C-domain active site) located in the N-terminal subdomain are represented as orange sticks.

**Figure 6 ijms-25-11774-f006:**
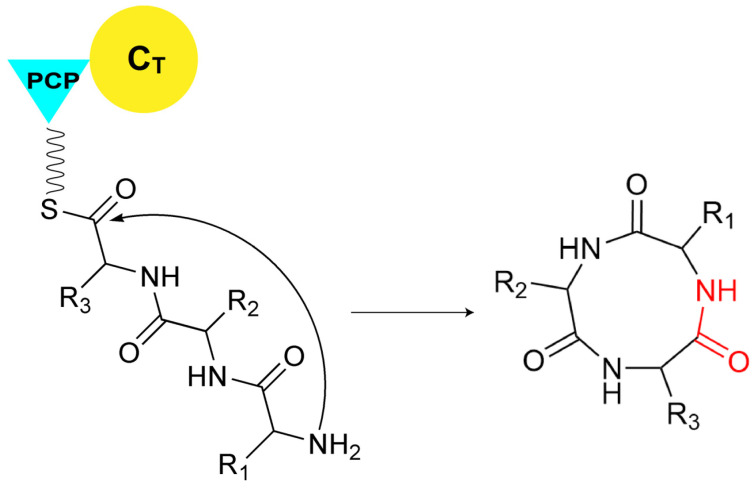
The C_T_ domain releases the final product through macrocyclization by employing the terminal amine as a nucleophilic agent.

**Figure 7 ijms-25-11774-f007:**
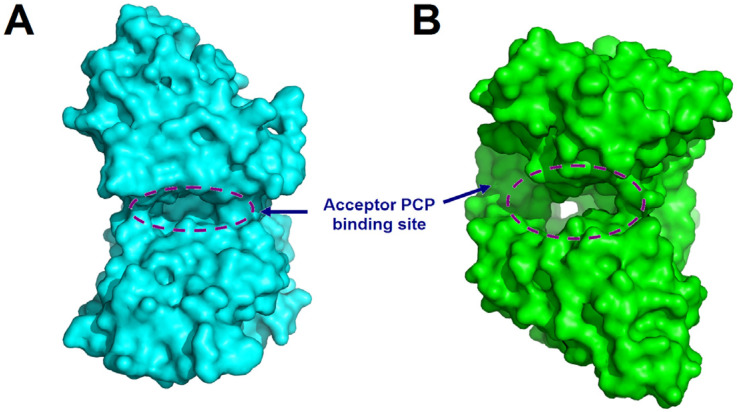
Structural comparison of the C_T_ domain of TqaA from *Penicillium aethiopicum* (PDB code: 5DIJ) (**A**) with canonical C domain (VibH) (**B**) illustrates the significantly narrower solvent channel and the blockage of the acceptor side in C_T_ domain. The structures are represented as surfaces, the regions corresponding to solvent channels are indicated by dashed circles, and the acceptor PCP domain binding sites are indicated.

**Figure 8 ijms-25-11774-f008:**
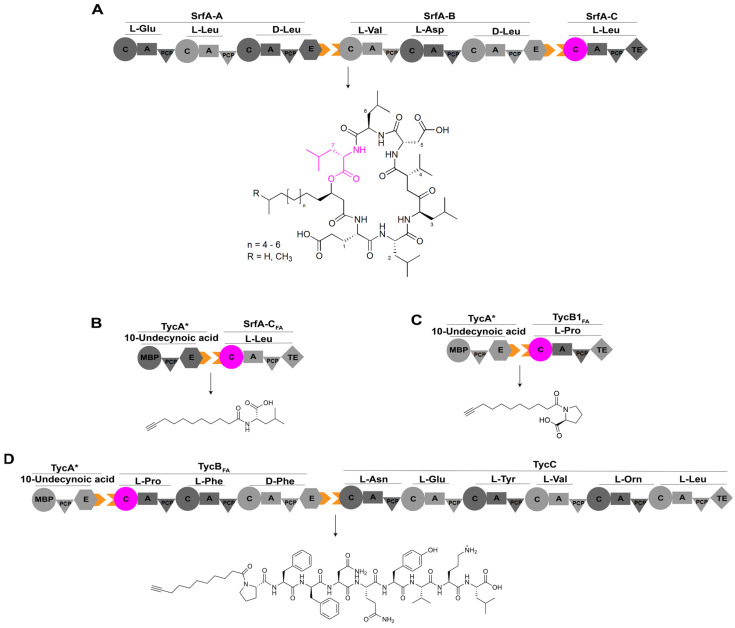
The assembly line of lipopeptide antibiotic surfactin and its engineering. (**A**) The assembly line of surfactin consists of three modules, SrfA-A, SrfA-B, and SrfA-C, contributing to the production of the cyclic heptapeptide. (**B**) Interaction of TycA* with SrfA-C_FA_ generates a novel product, 10-undecynoyl-L-Leu. (**C**) Interaction of TycA* with TycB1_FA_ yields a novel product, 10-undecynoyl-L-Pro. (**D**) TycA*, TycB_FA_,and TycC combine to produce a novel lipopeptide, 10-undecynoyl-L-Pro-L-Phe-D-Phe-L-Asn-L-Gln-L-Tyr-L-Val-L-Orn-L-Leu. Abbreviations: TycA*: variant of TycA with its A domain replaced by maltose-binding protein (MBP); SrfA-C_FA_: variant of SrfA-C harboring mutations W143T, Y145V, and F155I; TycB1_FA_ and TycB_FA_: hybrids with C domain of SrfA-C_FA_ (residues Gln10-Gln430) inserted into the C domain of a standalone elongation module TycB1 and the first C domain of full-length TycB responsible for tyrocidine synthesis, respectively; Orn: ornithine.

**Figure 9 ijms-25-11774-f009:**
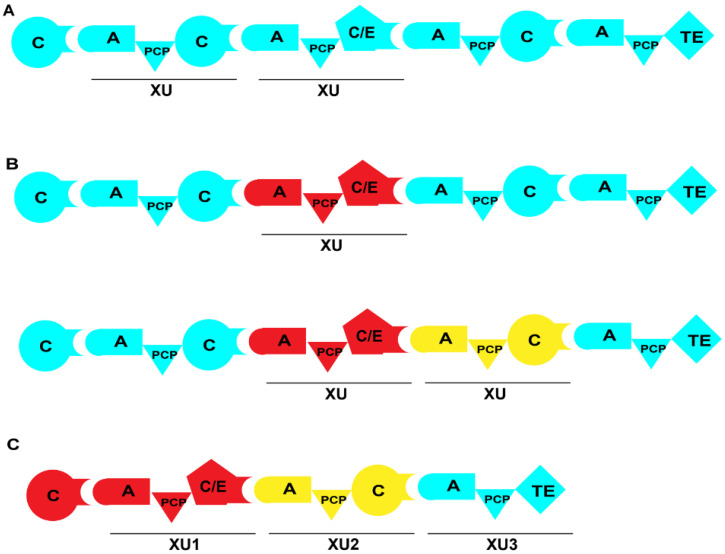
Strategy of NRPS engineering involving XU. (**A**) Typical XUs within the NRPS module. (**B**) Recombinant NRPS with the replacement of one or multiple XUs. (**C**) De novo construction of NRPS by using XUs. XUs from different NRPSs are represented in various colors.

**Figure 10 ijms-25-11774-f010:**
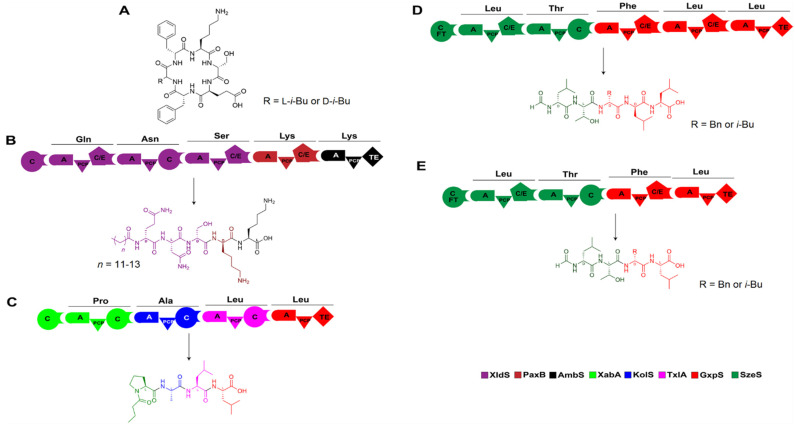
Artificial construction of NRPSs by XUs and the corresponding products. (**A**) Structures of two ambactin derivatives generated by domain and module swapping of the ambactin-producing NRPS AmbS. (**B**–**E**) Several non-natural peptides synthesized de novo using XUs from various sources, which are also listed in Table 1. Abbreviation: FT: formyl transferase.

**Figure 11 ijms-25-11774-f011:**
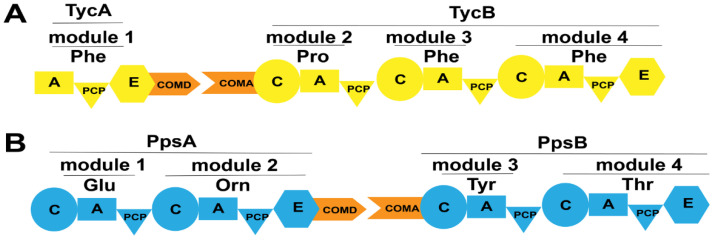
The location of COM domains in several NRPSs. (**A**) The COM domains between the TycA and TycB subunits in the tyrocidine assembly line; (**B**) The COM domains between the PpsA and PpsB subunits in the plipastatin assembly line. Abbreviations: COMD: the donor COM domain; COMA: the acceptor COM domain.

**Figure 12 ijms-25-11774-f012:**
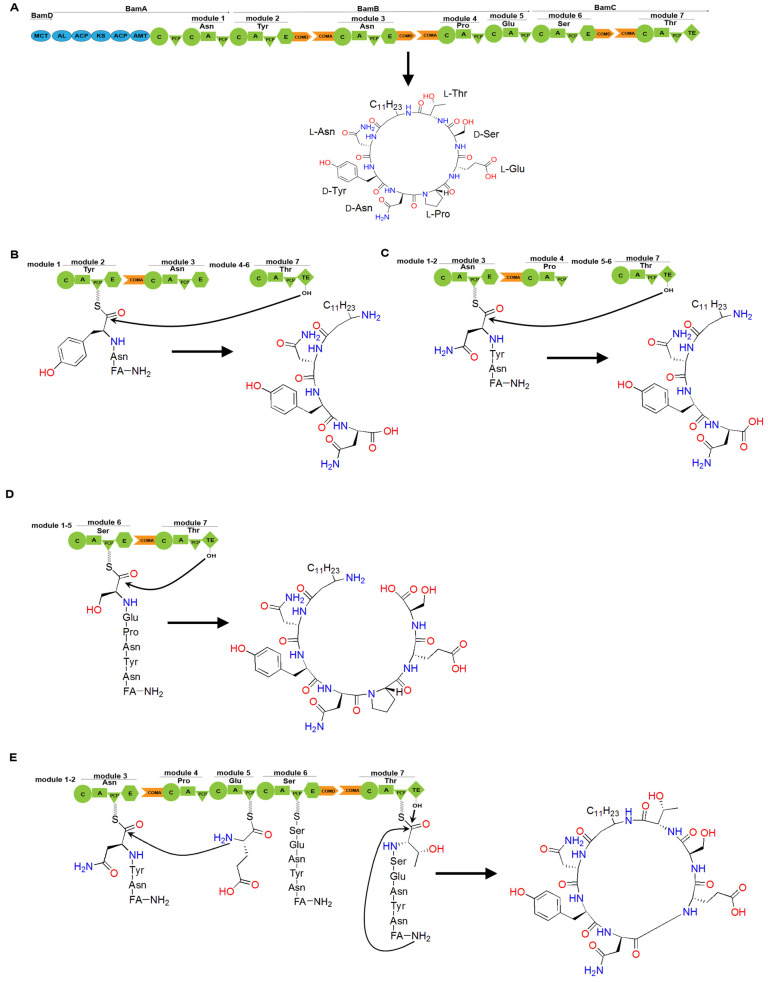
The assembly line of bacillomycin D and its engineering. (**A**) The cyclic lipopeptide structure of bacillomycin D is composed of a β-amino fatty acid chain linked to a heptapeptide. Bacillomycin D synthase is a PKS/NRPS hybrid enzyme comprising four units: BamD, BamA, BamB, and BamC. The blue regions in bamD and bamA represent PKS domains. (**B**) Deletion of the COMD domain in module 2 results in premature hydrolysis by the TE domain, generating a linear dipeptide linked to a β-amino fatty acid chain. (**C**) Deletion of the COMD domain in module 3 leads to premature hydrolysis by the TE domain, generating a linear tripeptide linked to a β-amino fatty acid chain. (**D**) Deletion of the COMD domain in module 6 leads to premature hydrolysis by the TE domain, generating a linear hexapeptide linked to the β-amino fatty acid chain. (**E**) Deletion of the COMD domain in module 3 leads to the skipping of module 4, generating a cyclic hexapeptide linked to a β-amino fatty acid chain. Abbreviations: FA stands for fatty acid.

**Table 1 ijms-25-11774-t001:** Details of representative cases of NRPS engineering by XU replacement.

Engineering Methods, Targets, and Purposes	XU Source	Number of XUs	Production of Peptide Product ± Standard Deviation (mg/L)	% of Wild-Type Level
Domain swapping	GameXPeptide-producing NRPS GxpS	1	0.31 ± 0.15	88
		2	0.20 ± 0.12	57
		2	0.52 ± 0.31	148
Recombination of NRPSs	GxpS KolS	3	16.12 ± 3.89	48.6
		4	15.82 ± 3.06	47.7
	XtpS AmbS	2	8.94 ± 0.73	28.1
			2.73 ± 0.07	8.6
	XtpS GarS	2	5.15 ± 1.59	16.2
			0.81 ± 0.18	2.5
	XtpS AmbS GarS	3	1.08 ± 0.32	3.4
			0.30 ± 0.18	0.9
	XtpS GarS GrsB	3	9.53 ± 4.07	30.0
			0.45 ± 0.27	1.4
			1.48 ± 1.05	4.7
	BicA	1	24.45 ± 3.82	77.0
			6.44 ± 1.34	20.2
De novo design of NRPSs	XldS PaxB AmbS	5	10.02 ± 0.44	-
			27.46 ± 0.65	-
			7.17 ± 0.22	-
	XabA KolS TxlA GxpS	4	7.53 ± 0.51	-
	SzeS GxpS	5	13.64 ± 0.43	-
			2.44 ± 0.03	-
	SzeS GxpS	4	30.60 ± 2.32	-
			4.57 ± 0.37	-

**Table 2 ijms-25-11774-t002:** Details of XfpSengineering involving COM domains.

Engineering Site	Engineering Method	Production of Peptide Product ± Standard Deviation (mg/L)	% of Wild-type Level
None	None	XFP A 1.21 ± 0.005	100
		XFP B 0.48 ± 0.03	100
C-terminus and N-terminus of XfpS	Introduction of Strep Tag II	XFP A 3.2 ± 0.6	265
		XFP B 1.2 ± 0.2	258
C-terminus of XfpS-1 and N-terminus of XfpS-23; N-terminus of XfpS-1 and C-terminus of XfpS-23	Introduction of exogenous COM domain; introduction of Strep Tag II	XFP A 4.5 ± 0.7	371
		XFP B 2.5 ± 0.6	526
C-terminus of XfpS-1 and N-terminus of XfpS-23; N-terminus of XfpS-1 and C-terminus of XfpS-23	Deletion of the COMD portion in the exogenous COM domain; introduction of Strep Tag II	XFP A 3.7 ± 0.3	308
		XFP B 2.5 ± 0.3	519
C-terminus of XfpS-1 and N-terminus of XfpS-23; N-terminus of XfpS-1 and C-terminus of XfpS-23	Deletion of the COMA portion in the exogenous COM domain; introduction of Strep Tag II	XFP A 0.83 ± 0.08	69
		XFP B 0.9 ± 0.1	178
C-terminus of XfpS-1 and N-terminus of XfpS-23; N-terminus of XfpS-1 and C-terminus of XfpS-23	Deletion of the COMD and COMA portions in the exogenous COM domain; introduction of Strep Tag II	XFP A 0.9 ± 0.1	73
		XFP B 0.84 ± 0.07	176
C-terminus of XfpS-12 and N-terminus of XfpS-3; N-terminus of XfpS-12 and C-terminus of XfpS-3	Introduction of exogenous COM domain; introduction of Strep Tag II	XFP A 0.06 ± 0.01	5
		XFP B 0.22 ± 0.02	46
C-terminus of XfpS-12 and N-terminus of XfpS-3; N-terminus of XfpS-12 and C-terminus of XfpS-3	Deletion of the COMD portion in the exogenous COM domain; introduction of Strep Tag II	XFP A -	-
		XFP B 0.04 ± 0.01	8
C-terminus of XfpS-1, N-terminus and C-terminus of XfpS-2 and N-terminus of XfpS-3	Introduction of exogenous COM domain	XFP A 0.11 ± 0.02	9
		XFP B 0.34 ± 0.03	71
C-terminus of XfpS-1, N-terminus and C-terminus of XfpS-2 and N-terminus of XfpS-3; N-terminus of XfpS-1 and C-terminus of XfpS-3	Introduction of exogenous COM domain; introduction of Strep Tag II	XFP A 0.37 ± 0.02	31
		XFP B 0.9 ± 0.1	195

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
