# Peer review of "Structure, Function and Engineering of the Nonribosomal Peptide Synthetase Condensation Domain"

_ijms, 2024, doi:10.3390/ijms252111774_

Round 1
Reviewer 1 Report
Comments and Suggestions for Authors
I am very grateful for having a chance to review this manuscript. This review represents a good overview of structure, function, and engineering of a condensation domain in non-ribosomal peptide synthetases. However, I feel that some major concerns should be resolved prior to publication.
Comments:
1. In this review article authors provide a good overview of the structure, function, and engineering of the C domain of NRPS, but it would be helpful to include more specific details concerning certain structural features and catalytic mechanisms of the domain are highly desired because this constitutes one of the core features of the biosynthesis in NRPS.
2. This review may expand further on current debates and controversies on the substrate specificity and catalytic mechanism of the catalytic domain.
3. Engineering NRPS systems through targeting the C-domain to produce a variety of derivative peptide natural products would be better presented with more examples and some relevant case studies to show specific successes and challenges in such an approach. Additionally, the review can mention some of the strategies and design principles which have been used in such engineering efforts.
4. The review presents the identification of new discoveries involving noncanonical C domains, several other C superfamily domains, which extended our knowledge on the C domain. The description of the discovery of novel C-domain variants and their structural and functional properties along with the putative applications in NRPS engineering is of greater value.
5. In general, the present review draws emphasis toward better understanding the mechanism of NRPS as an entity for making more efficient and stable engineering strategies. In this respect, discussion of the current state of knowledge regarding the broader mechanism of NRPS and its relevance for application in enhancing the C domain engineering would be of value.
6. Regarding the figures, this article may include figures that compare the structural and functional features of the noncanonical C domains with other previously found C superfamily domains. This may help the readers to appreciate the range of diversity and versatility C domain-like enzymes can exhibit.
7. In this review author could include conclusion with a figure or schematic overviewing the larger mechanism of NRPS and how C domain fits into the biosynthetic process which can give contextual meaning to the importance of C domain engineering within the larger NRPS framework.
8. This review could include the limitations regarding the present knowledge on mechanisms of C domains and challenges that come up with NRPS engineering could have been more extensively discussed for balance in the field.
Author Response
Response to Reviewer 1
We thank the reviewers for their careful read and thoughtful comments, which could significantly improve the quality of our manuscript. We have carefully taken their comments into consideration in preparing our revision and have responded to each point raised in the review. Details of the revisions made to the manuscript in response to these comments are listed below. All the changes to the original manuscript have been marked in blue in the revised manuscript.
Point 1: In this review article authors provide a good overview of the structure, function, and engineering of the C domain of NRPS, but it would be helpful to include more specific details concerning certain structural features and catalytic mechanisms of the domain are highly desired because this constitutes one of the core features of the biosynthesis in NRPS.
Response 1: Thanks so much for the positive evaluation and the proposed disadvantage of our manuscript. After carefully surveying the literatures, the structural features and catalytic mechanisms of the C domain have been further elaborated in the revised Section 2.
Point 2: This review may expand further on current debates and controversies on the substrate specificity and catalytic mechanism of the catalytic domain.
Response 2: Thanks so much for this suggestion. After carefully surveying the literatures, we observed that, apart from the C domain, there are no major debates or controversies regarding the substrate specificities and catalytic mechanisms of the other core catalytic domains in NRPS. The existing debates and controversies on the substrate specificity and catalytic mechanism of the C domain have been comprehensively discussed in the revised manuscript.
Point 3: Engineering NRPS systems through targeting the C-domain to produce a variety of derivative peptide natural products would be better presented with more examples and some relevant case studies to show specific successes and challenges in such an approach. Additionally, the review can mention some of the strategies and design principles which have been used in such engineering efforts.
Response 3: Thanks so much for this suggestion. Several new examples and case studies of NRPS engineering relevant to the C domain, including Cs domain engineering, exchange of a single C domain, and C/E domain relocation, have been added in the revised Sections 4.2 and 4.3. Meanwhile, the strategies, design principles and prospect of these engineering methods have been described in revised Sections 4.2 and 4.3 and 5.
Point 4: The review presents the identification of new discoveries involving noncanonical C domains, several other C superfamily domains, which extended our knowledge on the C domain. The description of the discovery of novel C-domain variants and their structural and functional properties along with the putative applications in NRPS engineering is of greater value.
Response 4: Thanks so much for this suggestion. The discoveries of these noncanonical C domains have been described in the revised Section 3. The structural and functional properties of these noncanonical C domains that are distinct from canonical C domains are described as well. NRPS engineering case studies relevant to several types of noncanonical C domains, such as Cs domain and E domain, have also illustrated in the revised Section 4.
Point 5: In general, the present review draws emphasis toward better understanding the mechanism of NRPS as an entity for making more efficient and stable engineering strategies. In this respect, discussion of the current state of knowledge regarding the broader mechanism of NRPS and its relevance for application in enhancing the C domain engineering would be of value.
Response 5: Thanks so much for this suggestion. The mechanism of iterative NRPS is introduced and a NRPS engineering case study of combining linear and iterative NRPS has been discussed in the revised Section 4.2, and the potential for advancements of NRPS engineering development taking advantages of NRPSs employing iterative, translocation, and module jumping modes, has been included in the revised Sections 5.
Point 6: Regarding the figures, this article may include figures that compare the structural and functional features of the noncanonical C domains with other previously found C superfamily domains. This may help the readers to appreciate the range of diversity and versatility C domain-like enzymes can exhibit.
Response 6: Thanks so much for this suggestion. As the Cy and CT domains possess the most significant features compared to the canonical C domain, two structural figures depicting the unique active site motif of Cy domain and the solvent channel in CT domain with the comparison to those of canonical C domain have been included in the manuscript. These graphic representations can help the readers to rapidly observe the distinct features of these noncanonical C domains.
Point 7: In this review author could include conclusion with a figure or schematic overviewing the larger mechanism of NRPS and how C domain fits into the biosynthetic process which can give contextual meaning to the importance of C domain engineering within the larger NRPS framework.
Response 7: Thanks so much for this suggestion. The major role of C domain involving in the NRPS assemble line have been depicted in the revised Figure2, which demonstrates the central role of C domain in peptide chain elongation.
Point 8: This review could include the limitations regarding the present knowledge on mechanisms of C domains and challenges that come up with NRPS engineering could have been more extensively discussed for balance in the field.
Response 8: Thanks so much for this suggestion. The main limitations of the current understanding of the mechanisms of C domains are the exact role of the second histidine residue within the conserved active site motif "HHXXXDG" and the general rules governing substrate specificity, which have been extensively described in the Section 2.2, 4.1 and 5. In addition, a more detailed discussion of the challenges in NRPS engineering relevant to these limitations and prospect in NRPS engineering have been included in the revised Sections 5.
Reviewer 2 Report
Comments and Suggestions for Authors
This is a very nice review article. It is a bit non-comprehensive, meaning that there is a lot more papers and work that could have been included.
but nevertheless the topis is nicelly covered.
I have two minor issue regarding graphical representations that are stated in the pdf

Author Response
Response to Reviewer 2
We thank he reviewers for their careful read and thoughtful comments, which could significantly improve the quality of our manuscript. We have carefully taken their comments into consideration in preparing our revision and have responded to each point raised in the review. Details of the revisions made to the manuscript in response to these comments are listed below. All the changes to the original manuscript have been marked in blue in the revised manuscript.
Two minor issues regarding graphical representations:
Point 1: Line28: picture or a graphical presentation of structures should be added.
Response 1: Thanks so much for this suggestion. A structural presentation of NRP modifications has been added as the revised Figure1.
Point 2: Figure5A: it would be nice to have slightly bigger structures.
Response 2: We apologize for the unclear presentation in the previous Fig5A and thank for this suggestion. The structure in that figure has been properly enlarged, which is suitable for the readers to clearly see the details of this cyclic structure.